# Family Involvement Training for Staff and Family Caregivers: Case Report on Program Design and Mixed Methods Evaluation

**DOI:** 10.3390/healthcare12050523

**Published:** 2024-02-22

**Authors:** Cecilia Marion, Shazmin Manji, Linda Podlosky, Heather MacGillivray, Tanya L’Heureux, Sharon Anderson, Jasneet Parmar

**Affiliations:** 1Covenant Health Canada, Youville Home, St. Albert, AB T8N 1K1, Canada; cecilia.marion@covenanthealth.ca (C.M.); shazmin.manji@covenanthealth.ca (S.M.); heather.macgillivray@covenanthealth.ca (H.M.); 2Family Caregiver, University of Alberta, Edmonton, AB T6G 2T4, Canada; lrrose@telus.net; 3Department of Family Medicine, University of Alberta, Edmonton, AB T6G 2T4, Canada; trpankiw@ualberta.ca (T.L.); jasneet.parmar@albertahealthservices.ca (J.P.)

**Keywords:** long-term care, family involvement, nursing homes, family caregiving, informal care, family caregivers, residential long-term care, quality of life

## Abstract

The COVID-19 pandemic underscored the imperative for meaningful family involvement in long-term care, aligning with policy and safety standards while enhancing outcomes for caregivers, residents, and staff. The objectives of this article are as follows: (1) a case study report on implementing a family involvement intervention designed to facilitate the formal and safe engagement of family caregivers in resident care and (2) the pilot evaluation of the intervention. We used Knapp’s six-step implementation science model to guide and describe intervention development to provide insight for others planning family involvement projects. We employed sequential mixed methods, including surveys with quantitative and qualitative questions before and after program implementation for providers, and surveys and interviews with family caregivers a year after. We used the Mann–Whitney U test (*p* < 0.05) to assess differences in health providers’ perceptions pre- and post-education. Families and staff perceived that the Family Involvement Program was important for improving the quality of care, residents’ quality of life and family/staff relationships. Providers’ perceptions of the program’s positive impact on residents’ quality of life (*p* = 0.020) and quality of care (*p* = 0.010), along with their satisfaction with working relationships with families (*p* = 0.039), improved significantly after the program. Qualitative data confirmed improvements in family–staff relationships. In conclusion, we documented the design of this family involvement initiative to encourage family caregivers and staff to work together in residents’ care. Youville’s Family Involvement Program gives families and family caregivers an explicit role as partners in long-term care. The mixed methods pilot evaluation documented improvements in staff and family relationships.

## 1. Introduction

Family caregivers’ role in long-term care is ambiguous [1,2,3]. A family caregiver is a chosen family member, friend, or neighbor who provides unpaid support, assistance, and care to someone who has a mental or physical illness, disability, or frailty and is unable to fully care for themselves. When the person needing care resides in the community, family caregivers provide 75 to 90% of the care [4]. After the care-receiver is admitted to a long-term care program, the assumption is that paid staff provide the care. Then, family caregivers find themselves navigating a complex landscape where their duties may not be clearly defined or acknowledged by formal care systems [5,6].

The residential long-term sector in Canada is regulated tightly by provincial and territorial governments with responsibility for healthcare [5,7]. In Canada, however, policies about family involvement were vague prior to the COVID-19 pandemic [5,7,8]. Typically, conflict between staff and caregivers can be traced to the lack of formal policies on family caregivers’ roles and responsibilities [5,7,9]. The pandemic underscored the need for meaningful family involvement in long-term care that aligns with policy and safety standards while enhancing outcomes for caregivers, residents, and staff [5,6].

The objectives of this research project were to (1) document the design of the Family Involvement Project to facilitate the formal and safe involvement of family caregivers in the resident’s care and (2) to report the pilot evaluation of the intervention. Because the structure of the ideal design and implementation of family involvement in long-term care is still limited [10], we used Knapp’s [11,12] six-step implementation science model to explicitly describe the design of the Family Involvement Program.

### Literature Review

Family involvement is the active participation of and support provided by family members regarding the care and well-being of the person who requires ongoing assistance and support because of frailty, illness, or disability. This involvement can take various forms, including emotional support, financial support, decision-making, and physical caregiving [3,13,14,15,16]. Gaugler [13] conducted a critical review in 2005 to understand if and how family involvement could be effectively integrated into long-term care to enhance residents’ quality of life, family caregiver well-being, and staff job satisfaction. He noted that extant research mainly focused on refuting assumptions that families abandon long-term cared for residents. Of the three interventions reviewed, the most effective incorporated family and staff in collaborative partnerships [13]. He also recommended designing family involvement interventions that de-emphasize medicalized, task-based philosophies to focus on homelike environments.

In their 2022 update, Gaugler and Mitchell noted more frequent family visits were associated with fewer psychological disturbances for residents with dementia and better resident quality of life [14]. How family involvement is optimized in assisted living and long-term care settings remains an ongoing research gap. Aligning with Gaugler and Mitchell’s findings, Hayward and colleagues [10] also reported that few interventions specifically promoted family involvement in the long-term care of residents with dementia. Notably, all of the interventions they reviewed reported positive results, including improvements in family and staff communication, families’ knowledge about dementia, and family participation. The impacts on residents’ quality of life, however, were mixed (positive and negative). The authors of both reviews [10,14] recommended research on ways to involve family caregivers in long-term care settings.

Research on the impacts of the COVID-19 pandemic also advanced the need to improve family involvement in long-term care [1,16,17]. COVID-19 restrictions focused on infection control, rather than on person and family-centered care or residents’ and family caregivers’ quality of life [5,14,18]. At the outset of the pandemic, family involvement was restricted [6,9,19]. Family caregivers and families were called visitors and unable to enter congregate care settings [20,21]. The lack of social interaction and COVID-19 policies such as restricting residents to their rooms, eating meals alone in their rooms, and mask wearing increased challenging behaviors in people with dementia [10,21]. Restrictions hastened residents’ physical and cognitive decline [21,22]. Family caregivers experienced loneliness [21,23], guilt, and anxiety at the irreplaceable lost time together [1,24]. As these impacts of COVID-19 restrictions emerged, one or two essential family caregivers were allowed into residents’ rooms to provide some care [20,21]. However, even this somewhat relaxed public health measure disrupted typical family care routines, particularly when three or more family members had been providing care [14,21,25]. The strict infection control measures limited family visits. The staff shortages that existed before the pandemic were exacerbated by COVID-19. Infection control measures and short staffing hindered the social and emotional connections between residents, families, caregivers, and staff [10,14]. Efforts to create a homelike environment in long-term care were disrupted.

COVID-19 family presence policies overlooked the amount of care that family caregivers provide in long-term care. It took time for policy makers and leaders to recognize the extent to which restrictions on family involvement increased healthcare providers’ workload. Coe and Werner’s analysis of population health survey data found that family caregivers of long-term care residents contributed 37.4 h of care per month on average, which is roughly equivalent to an extra full-time shift of paid providers per month [16]. The long-term care staffing shortages that existed prior to the pandemic [8,24] were exacerbated by restrictions on what family caregivers could do [17], leading to increased work to follow COVID-19 policies [23,25] and trying to meet residents’ increased care and socioemotional needs [16,17] while supporting family visitations through window visits or with technology such as iPads or ZOOM [6,26,27]. The pandemic reinforced that family involvement in long-term care is essential for promoting person-centered care, improving the well-being of residents, and fostering a collaborative approach to caregiving [5,10,14]. Reviewers, however, charge that the science of promoting family involvement in long-term care is limited to caregiving [10,14]. Beyond family council meetings, few family involvement programs which allow for constructive feedback while avoiding conflict between family and staff have been piloted [13,28].

The extant research does offer guidance on improving family involvement in long-term care. Hayward and colleagues [10] suggest that policy change is not sufficient to improve family involvement. They advise that interventions specially designed to improve family involvement are required [10,14,23]. Further, interventions need to be planned with an understanding of the system policies and organizational processes that enhance or impede meaningful family involvement [5,29]. Reviewers also suggest that family caregivers should be recognized as key partners in the design and their perspectives should be integrated into the decision-making processes. In reports, the invention design and implementation plans need to be provided in more detail [28]. More comprehensive descriptions of program development should improve the replication of family involvement initiatives [28].

In what follows, we describe the setting and then the design and implementation of the Family Involvement Program using Knapp’s [11,12] ADAPT (assessment, deliverables, activate, pre-training, training, and sustainability) implementation science framework to explicitly describe the design of the Family Involvement Program. This case report description should help others planning family involvement projects to foresee what might be involved. Then, we present the results of our mixed methods pilot testing.

## 2. Family Involvement Program Design

Youville Home is a long-term care facility in Alberta. It includes the original nursing home built in 1963 and a newer facility that opened in 2006. In the province of Alberta, long-term care spaces are reserved for individuals who have highly complex and unpredictable health needs who require 24 h onsite registered nurse assessment and/or treatment. Long-term care residents have medical and physical care needs that cannot be safely provided in their own home or in supported living. People working in long-term care refer to it as the resident’s end-of-life home [30].

### 2.1. The Healthcare Team

Staffing comprises community physicians who come into the site to provide care to residents, a nurse practitioner, registered nurses, licensed practical nurses, and healthcare aides. In addition, physical therapists, occupational therapists, recreational therapists, dietitians, spiritual carers, and social workers complete the staffing. This staffing mix is the current practice in Alberta.

### 2.2. Residents

Long-term care residents have very complex care needs [31]. In Alberta, long-term care facilities are designated for individuals with serious fluctuations in health status requiring immediate health professional assessment; a need for medication management and other treatments; conditions requiring the continued presence of a registered nurse and the consultative availability of rehabilitation or dietary professionals; unpredictable behavior that places the individual or others at risk; and complex end-of-life care needs whose care cannot be safely provided in their own home or in supported living (https://www.albertahealthservices.ca/cc/Page15502.aspx accessed on 30 January 2024). Within Youville Home, during the study period, there were 232 residents with an average age of 81.5 years. About 40% of residents had a diagnosis of dementia and/or Alzheimer’s. The average length of stay was 428.74 days.

### 2.3. Assessment: The Project Design

The objective in this project was to make sure that family members who wanted to help care for the resident that they cared for followed best practice to ensure positive experiences, outcomes, and safety for residents, family members, and staff. Typically, when residents are admitted to long-term care, the care provided follows a medical model of care. Residents admitted to long-term care come from acute care, home, and assisted living because they require nursing care 24 h a day. The assumption has been that nursing home staff take over all of the care and that families relinquish much of their involvement and responsibilities involved in the care that they had been providing up to admission. In this medicalized model, families are assumed to become visitors who adapt to the practices and routines of the nursing home. However, family caregivers have challenged the assumption that they were just visitors. They have advocated recognition of their role and to be engaged in caregiving decision-making processes in long-term care settings.

In 2016, a core team of management and leaders from Youville Home and researchers gathered to address this gap. The envisioned Family Involvement Project in Youville could be “A collaborative and vibrant community where families are supported to participate at their desired level of involvement, and to create options to enhance the care provided to each of our unique residents”. They explicitly set out to establish processes and policies that facilitated and evaluated family involvement in residents’ support and care. The hope was that they could design a family involvement program that would help long-term care leaders to shift their perception from family members as visitors to families as partners in care. They began by examining the risks, how policies might influence family involvement, and potential project timelines. Youville leaders prepared a high-level workplan and submitted a funding application (2018). Funding was approved on 3 August 2018. The project received research ethics approval from Covenant Health and the University of Alberta Health Ethics Research Board (Pro00091450).

### 2.4. Deliverables: Creating the Family Involvement Processes

Long-term care policies and practices are set in traditional medical and nursing models, so we assumed the role change from family as visitors to partners in care would require changes in culture and practice. Our team acknowledged that we were designing an initiative that would re-shape the mindset of staff, families, and residents toward increased family involvement in resident care, which we expected to strengthen continuity of care and improve the relationships between the family, staff, and residents (Youville Home Family Involvement Program handbook, 2016, p. 2 [32]). At its core, the project involves reclaiming the central role that families have historically and rightly taken in the care relationship while ensuring an appropriate balance between the family’s willingness, capacity, and competency to provide care without unduly burdening them or putting them or the resident at risk of harm. We also wanted family involvement to support the continuation of natural relationships and care in our long-term care setting. To us, this meant that families would be offered opportunities to partner with care staff wherever reasonably appropriate to care for residents’ spiritual, emotional, social, and physical needs holistically. The project’s goals included (1) maintaining a natural relationship and level of care that was there before the resident moved into Youville Home and (2) ensuring that the resident and their family are partners with the care team.

#### 2.4.1. Steering Committee and Workplan

The Youville Home Oversight Committee, the Steering Committee, and six subcommittees were formed. A detailed project workplan was created. See Table 1 for the workplan sequence.

#### 2.4.2. Oversight Committee

The Oversight Committee had the responsibility of planning the project, guiding development of the Project Charter, and providing direction and oversight to ensure the project stayed within scope, on schedule, and within budget. All subcommittees reported to the Oversight Committee. See Table 2: committees and roles.

Knowing that relationships with family members and/or friends are an integral part of the care setting, family members were integral members of all committees from the outset. The Resident and Family Council was also informed, and we sought their feedback throughout the project life cycle.

#### 2.4.3. Considering Roles, Care Opportunities, Risks, and Policies

In order to identify needs and priorities and to determine subjective feelings about family involvement in residents’ care, we conducted surveys and interviews with families (n = 32), residents (n = 10), and staff (n = 32). Three quarters (76%) of respondents in the three groups agreed that strategies or initiatives were needed to involve families. Residents were most supportive, as 100% of the residents wanted families involved. There were a broad range of suggestions about how to involve families, from asking families verbally to personalizing letters asking families how they wanted to be involved.

Before considering involving families in care opportunities, the Policy and Standards Subcommittee looked at the policies, regulations, and legal considerations of involving families in the care provided in a long-term care setting. Work was necessary to make policy changes, educate staff on these changes, and support (1) families regarding safety and best practices in providing care to residents, (2) staff on partnering with families providing care, and (3) practices for resident consent. Then, the project was presented to the Facility Living Operators and the Continuing Care Medical Advisory Committee, which includes facility living directors and physicians providing continuing care. They noted the value of bringing families in and supporting them to help care for their family members.

The next step was consulting legal services to ensure that we could comply with relevant laws and regulations related to healthcare, patient privacy and rights, and family caregiver rights. To address potential risks, concerns, or conflicts relating to the impacts of involving families on healthcare workers’ roles or responsibilities, we reached out to professional colleges, associations, and staff unions for their cooperation and input.

The Roles and Responsibilities Committee looked at all the care opportunities, then worked with the Risk Management Committee to determine the safety risk of each activity to residents, staff, and family caregivers. See Figure 1: care opportunities.

The next step was to create a process for the implementation of family involvement into the general practices at the site. Legal services recommended that family caregivers who wished to be involved in high-risk care opportunities be registered as volunteers and then take volunteer training. As there was no predefined definition of “family involvement volunteer” in our mandate, volunteer services collaborated with us to develop the definition and role description. Once approved by volunteer services, the family involvement volunteer role was vetted through the Provincial Volunteers Committee. Volunteers are in a position of trust in their role with vulnerable older people and may develop a close, personal bond. Volunteers complete an application, police check, interview, basic health screening, and orientation and training relevant to their role. Family members who chose a high-risk care opportunity enrolled as a volunteer, then went through the security and orientation processes, similar to other volunteers.

#### 2.4.4. Roles and Responsibilities

The Roles and Responsibility Subcommittee then delineated the referral and documentation processes and pathways to ensure that caregivers could successfully and efficiently move from talking to the Resident Care Manager about care opportunities to enrolling as a volunteer, receiving the training, and assisting with care. See Figure 2: referral process. This committee also developed a conflict resolution process to address the times when staff and caregivers may struggle with maintaining a trusting relationship.

### 2.5. Activation and Pre-Training: Developing the Education

Our Education Subcommittee worked with the Steering Committee and families to create an education curriculum for family caregivers based on care opportunities. The educator also ensured that instructions and processes were laid out for families partnering with staff and staff partnering with family caregivers. Then, the steering and Education Subcommittees decided on the instructional materials needed, after which the Production Committee worked on scripts and managed production for the instructional videos. Students from the Northern Alberta Institute of Technology Digital Media and IT program worked on the video production and editing. Eight videos were produced. We used a flipped classroom educational approach where families would watch the instructional videos and then participate in a demonstration back approach with a staff educator. To make the videos accessible to families, family caregivers could use the Family Involvement iPad to view the videos when they were at the bedside, and they were also placed on YouTube.

We talked to staff and created surveys on the education they thought might be needed regarding the Family Involvement Program. Staff asked a lot of questions, including about the time constraints, responsibility in adverse events, identifying struggling caregivers, and working with families who have high expectations. Data were compiled and either added to staff education or added to risk mitigation strategies in the care opportunities protocols. For example, family caregivers could assist with meals in the dining room rather than in residents’ rooms, so staff training in managing choking and CPR were available to assist. We also created a frequently asked questions page as part of the staff education and a conflict resolution tip sheet for both family caregivers and staff. As we required documentation of who did what and when, new family involvement records were programmed into the PointClickCare records system. The Communications Committee vetted all of the materials to ensure that they were clearly written and understandable.

### 2.6. Training: Pilot Testing

Given that planning for the project began before the World Health Organization declared the COVID-19 pandemic in March 2020 and site access was restricted, the detailed project workplan, devised and approved at inception, was revised as site access reopened. Pilot testing began in 2022.

### 2.7. Sustainability

The Sustainability Committee managed the processes, standards, and strategies designed to sustain the program. This included ensuring that effective change management was in place and ensuring that the program continued to meet the Continuing Care Health Service standards and Alberta Accommodation standards for long-term care, and that Youville Home procedures and policies were followed for sustainability. It also included processes that would ensure the Family Involvement Program became general best practice at the site.

## 3. Methods: Evaluation of the Family Involvement

Like most long-term care settings in Canada, the Youville Home continues to cope with periodic COVID-19 outbreaks, increased workloads, and short staffing. Thus, we developed a less onerous pilot evaluation approach to assess the practicality of the Family Involvement Program. Given the aforementioned stressors, we chose a convergent mixed methods design to gain as full an understanding as possible. Both quantitative and qualitative data were collected together, and then in an analysis, we compared the quantitative and qualitative data to see if the data either support or contradict each other [33]. To assess the practicality of the pilot Family Involvement Program, we used a short survey for family caregivers one year after the education program was implemented and before the education was implemented and one year later for healthcare providers. The survey questions for family caregivers included a question about their relationship to the resident, if they were aware of the Family Involvement Program, and Likert scale questions on the importance of being involved in their resident’s care, satisfaction with the level of involvement in resident’s care, and the program’s importance in improving residents’ quality of life, quality of care, and family/staff relationships. We used repeated measure surveys for providers in 2022 before the program began and 1 year after implementation in 2023. Questions included perceptions of family involvement before the program began and providers’ and family caregivers’ observations a year later. We asked one demographic question about their roles, if they received education about the project, two questions about their satisfaction with the education, and Likert scale questions on the program’s importance in improving residents’ quality of life, quality of care, and family/staff relationships that mirrored those asked of family caregivers. We also asked a Likert scale question on their satisfaction with their relationship with family caregivers. In both family caregivers’ and providers’ surveys, we included “tell us more” qualitative questions about the impacts of the program on resident quality of life, quality of care, and family/staff relationships.

### 3.1. Participant Recruitment

Staff were provided with a survey pre-education on the nursing units to be completed anonymously. The surveys were collected in a collection box and the staff educator collected the surveys from the box. The receptionist provided the questionnaires to family caregivers to complete when they signed in to enter the building. A family caregiver questionnaire collection box was placed at the reception desk. Following the surveys, we conducted semi-structured interviews with family caregivers (n = 3) to further understand their view of taking the training and partnering with staff to provide care. See Appendix A for surveys and interview questions.

### 3.2. Data Analysis

Quantitative data were analyzed using IBM SPSS Statistics for Windows, version 26.0 (IBM Corp., Armonk, NY, USA). We used descriptive statistics to summarize the data. With the small sample size and non-normality of the data collected (Shapiro–Wilks test *p* < 0.05), we used the non-parametric Mann–Whitney *U* test (*p* < 0.05) to evaluate the difference in health providers’ attitudes before and after the education. The Mann–Whitney *U* test is the non-parametric alternative to the t-test for independent samples. It compares medians by converting the scores to ranks across the two groups, then evaluates whether the ranks for the two groups differ [34]. We used 0.05 as the significance level, conservatively reporting the two-tailed and equal variances not assumed in the data.

Qualitative interpretive description, outlined by Thorne [35,36] was our theoretical methodological approach. It is a pragmatic approach designed to embrace the complexity and contradiction of health studies. Thorne [36] describes it as “ways of thinking that acknowledge the messiness of the everyday practice world” (p. 29). Two researchers not connected to the long-term care setting used constant comparison inductive content analysis [37] to analyze the qualitative survey responses and family caregiver interview data. Content analysis can be inductive or deductive. Deductive content analysis creates categories from the data based on a theoretically driven matrix. We used an inductive approach, whereby we formulated codes from the data. Interviews were transcribed verbatim and reviewed for accuracy. The qualitative survey and interview data were imported into NVivo for ease of data management. The two researchers independently read the transcripts and then openly coded the data separately. Participants’ responses could have several different codes, and researchers sometimes coded data in two codes. They discussed the codes and came to a consensus on the themes.

## 4. Results

Before the Family Involvement Program was introduced, 15 health providers (12 healthcare aides, 1 comfort care aide, and 2 nurses) completed the survey. A year after implementation, 44 health providers (34 healthcare aides and 10 nurses) and 20 family caregivers (10 spouses and 10 sons/daughters) completed the short anonymous surveys. Three family caregivers were interviewed.

Prior to the intervention, most providers (64.3%) heard about the program from their supervisor, another staff member told four others (28.6%), one saw a poster, and one did not answer. A year later, most (17/44, 41.5%) providers found it by looking for educational opportunities. Almost a third heard about it from a supervisor (29.3%), six (14.6%) from staff email communications, another staff member told three others (7.3%), two heard about it in their onboarding orientation, a family caregiver told one, and three did not answer. At the time of the 2022 survey, only six of the fifteen (40%) providers who had completed the survey had completed the training on partnering with family caregivers. A year later, 41 of 44 (93.2%) providers completing the survey had been trained.

Most family caregivers (18/20, 80%) were aware of the Family Involvement Program and 16 of the 20 completing the survey had received training. All 16 who had taken the training felt supported by staff to carry out the care activities. Six were very satisfied and eleven were satisfied with their level of involvement in care. One person was not satisfied but had not taken the education. The person did not want to participate in hands-on caregiving because of caring for a spouse in Youville and an adult child in a group home. Nine caregivers had been given the opportunity for more training, eight reported that they had not been given the opportunity for other training, and six did not answer the question.

### 4.1. Ratings of the Family Involvement Program

A year after the Family Involvement Program began, both family caregivers and providers responding to the survey rated the program as important for improving resident’s quality of life (mean out of five, unimportant to very important: family caregivers 4.55; providers 4.54), and quality of residents’ care (family caregivers 4.5; providers 4.74). Family caregivers also thought that it improved family and staff relationships (mean 4.21/5) and they were satisfied with their level of involvement in care (mean 4.22/5).

Staff were significantly (*p* = 0.003) more satisfied with the family involvement education in 2023 after they had experience with the program (mean 4.45, SD 0.6) than they were in 2022 (mean 2.66, SD 1.6) when only a few providers had taken the education. See Table 3 for family caregivers’ and Table 4 for providers’ perceptions of the Family Involvement Program.

Using the conservative two-tailed significance values, the Mann–Whitney *U* test found a statistically significant improvement in health providers’ 2022 pre-program and 2023 post-program ratings of the family involvement education and family involvement in resident care improving residents’ quality of life’ (*p* = 0.020) and quality of care (*p* = 0.010). The staffs’ satisfaction with their working relationship with families also improved significantly (*p* = 0.039). See Table 4 for a comparison of providers’ experience in 2022 and 2023.

### 4.2. Qualitative Results

Families’ and providers’ written comments mirrored and expanded on the benefits of the program. “*We both want what is best for the resident*” was the overarching theme in family caregivers’ and health providers’ comments. Family caregivers indicated they felt residents were important to staff, “*I believe everyone is here to help the resident*”. Providers spoke about increased collaboration, “*Now we can build the relationships with the residents and family to give the resident better care*”. Following the qualitative questions asked in the surveys, family caregivers and providers wrote positive comments about the Family Involvement Program improving residents’ quality of life and quality of care, and strengthening staff/family caregiver relationships. In the interviews, the family caregivers also spoke encouragingly about the benefits.

#### 4.2.1. Improved Quality of Life and Care

As this quote from a family caregiver illustrates, families felt more welcome and that the collaboration improved the atmosphere of the unit, “*When we work together, it’s not just the resident who benefits, but it seems to bring joy to the entire unit*”. Providers also thought that better relationships with families improved the quality of care, “*It is important to consider the quality of care with the involvement of the family. This program provides a friendly environment for both family and staff to give care to the resident*”. Staff and families spoke of how more family involvement reduced residents’ loneliness and improved their mood, and in turn their quality of life: “*I think the Family Involvement Program is very important in improving residents QoL because most of the residents get lonely and want someone to talk to*”.

Families felt better equipped to contribute effectively to residents’ care. According to staff, family involvement in care resulted in residents receiving care more efficiently and in a timely manner, “*Family can help residents with simple tasks like when the HCAs are busy and residents don’t have to wait for too long*”. Families were particularly pleased to be able to assist with the mechanical lifts which require two operators. If the family assisted one staff member with the mechanical lift, the resident did not have to wait to get into bed or out of bed and into a wheelchair: “*It is so much better for everyone, when I can help with the lift. She gets to bed sooner. I helped, I feel good and I think the staff feel good too*”.

Both staff and families thought that family involvement improved individualized, person-centered care. After the education, staff noted that families knew residents best, and they valued that knowledge: “*Family members know them better/well. Family knows what they like and how they are when it comes to daily living/care*”. Families focused on the impact of assisting with resident’s unique needs: “*My involvement has meant that I can relay little preferences and nuances of my grandmother’s needs that the staff might not be immediately aware of*”.

#### 4.2.2. Strengthening Staff and Family Relationships

Before the implementation of the Family Involvement Program, like the low ratings of family/staff relationships on the Likert scale question, staff comments portrayed difficult relationships with families: “*They will say why we put her here was for us to take care of her*”. A year after program implementation, both staff and family caregivers noted that working together as a team improved communication and understanding of the other’s situation, and that the program “*Opened up communication between staff and families*”. Staff thought that family involvement increased families’ knowledge of their contributions and reduced their work load: “*Family is more understanding of what staff will do and why things take so long which built trust*”. Family caregivers noted that communication with staff shifted from presenting problems to staff to engaging with staff in giving care: “*Engaging with the staff not just about problems but about everyday things has made me realize how hard they work and how we can collaborate better*”. Family caregivers thought that they eased the burden on healthcare staff by filling in gaps, and being actively involved made them feel like partners, “*I used to feel like just another visitor but being actively involved has made me feel like part of a larger caregiving team*”.

Uniquely, staff noted that family involvement helps residents feel like they are living in their own home rather than in a long-term care setting, “*When they see their family members, they felt like it is home, and it changes their morale*”, and that improved relationships with families: “*Yes. The relationship between the two has mostly helped to improve the quality of care for the residents. Sometimes when staff are stressed and family are there to talk to staff, generally it helps relieve the stress*”.

Family caregivers only observed that long-term care was underfunded and worried that their involvement might discourage governments from properly funding healthcare staffing, “*I became aware of lack of sufficient health care dollars to adequately support Youville Home… Covid has revealed so many shortcomings of our long-term care system*”. Several stressed that family involvement is beneficial, but they did not want it to replace systematic improvements in healthcare staffing and funding, “*I am happy to help my loved one, but I would never want to feel that the staff are being reduced and family is expected to do more. I feel that as it is, staffing is not adequate*”.

#### 4.2.3. Recommendations for Improvement

Both staff and families suggested that families should be informed about the opportunities for family involvement by providing “*Information for the family, let them know about this program*”. A family member noted that it was advertised: “*I don’t know that a lot of people know about it, even though there are signs up, pictures up, brochures around*”.

As well as brochures, printed material, and newsletter stories advertising the program, both family caregivers and staff suggested introducing it to families at different times through ongoing conversations. Both groups recommended starting at the initial visits prior to admission, during the admission process, and later at family conferences. They also thought that families involved in the program could talk to new families.

Staff and family members provided suggestions to improve the content of the program. Two staff members thought that the education for providers could be simplified, “*Don’t make it so formal and long*”, and that you have “*A lot of info thrown at you*”. A family member wondered if the education could include information and strategies to make the transition from home to long-term care easier: “*I am curious about whether or not you might have any new information on how a home caregiver like myself, and a long-term care facility like Youville Home—how we can collaborate to make this transition easier and more acceptable for the families?*”.

## 5. Discussion

Planning, designing, and implementing this Family Involvement Program into the Youville long-term care setting was a complex endeavor. The COVID-19 pandemic, which started just as Youville was about to implement family involvement training for staff and to recruit families to participate, increased the complexity. The public health pandemic protocols disrupted efforts to create a homelike environment in long-term care by imposing strict infection control measures, limiting family visits, and causing staff shortages, all of which hindered the social and emotional connections between residents, families, family caregivers, and staff. It quickly reinforced the need to explore effective strategies for facilitating family engagement in long-term care, improving communication between families and staff, and addressing barriers to family involvement, such as infection control measures. In the discussion that follows, first, we discuss the design of the Family Involvement Program, then our evaluation, and finally, the strengths and limitations of the research.

Designing and implementing this program required the collaboration of multiple, multilevel stakeholders, from senior and site leadership, legal and regulatory advisers, to staff and families. This enabled ownership of the program and successful design and implementation by the nursing home administration and staff. In his 2005 review looking toward the next generation of Family Involvement Programs, Gaugler [28] reported that a significant barrier to implementing family involvement programs is nursing home administration and staff reluctance to “adopt” the intervention on their own. Gaugler recommended engagement with a range of multilevel stakeholder perspectives in program design, including senior leadership, legal, policy, communication, family, and resident. Our documentation of the planning and implementation process responds to reviewers’ recommendations for more comprehensive intervention descriptions to facilitate the replication of family involvement programs [10,13,38,39].

This Family Involvement Program formalized family caregivers’ role in long-term care. Family caregivers are asked what they might want to do from an array of activities. Training for family caregivers was comparable to staff training and delivered by the same staff who train staff. Staff were also educated on how to partner with family caregivers. Notably, the intervention design includes most of the key components of family-centered care found by Kokorelias et al.’s [40] scoping review: (1) collaboration between family members and healthcare providers, (2) consideration of family contexts, (3) policies and procedures, and (4) family and healthcare provider education. Person and family-centered care is a health system goal in Alberta and a hallmark of high-quality care. It also aligns with Tasseron-Dries and colleagues’ [41] best practice recommendations on involving family caregivers. They defined best practice as “a practice that successfully involves family caregivers, potentially inspiring healthcare professionals and family caregivers to optimize joint caregiving”. In our Family Involvement Project, both staff and family caregivers reported that working together facilitated more collaborative, partnered caregiving for residents.

The quantitative evaluation of the Family Involvement Program adds additional evidence that family involvement in long-term care can improve staff–family caregivers’ relationships. There were significant positive improvements in providers’ perceptions of residents’ quality of life, the quality of care, and their relationship with families. Hayward and colleagues [10], in their 2022 review of family-oriented interventions in long-term care, reported that all twenty-two interventions designed to promote family involvement were associated with better communication and family–staff relationships. The impacts on residents’ quality of life, however, were mixed. In our study, staff and family caregivers thought that working together improved staff/family relationships, which in turn improved residents’ quality of life.

Partnering enabled timely care to residents. Based on Song et al.’s [42] recent study, more responsive, less rushed care should reduce responsive behaviors for residents with dementia. Also, Song and colleagues [42] found that when care was rushed, staff experienced yelling and screaming, verbal threats, hurtful remarks or behaviors, and even being spit on, bitten, hit, pushed, or pinched. Timelier care that reduces responsive behaviors should improve the environment for healthcare aides. Improvements in residents’ behaviors and better relationships with families should increase staff satisfaction and retention. Aligned with Song’s work [42] and Kokorelias et al.’s [40] evaluation recommendations, further evaluation of this program should include measures of responsive behaviors and staff satisfaction.

This study has several strengths. Even though it was disrupted by the COVID-19 pandemic and ongoing outbreaks, the program planning and development was rigorous and has been well documented. Leadership and staff could implement the program in stressful circumstances. Collaborative, co-design work like this, which includes engaged staff, families, site leadership, senior leadership, as well as communication, legal, and volunteer leaders, will enable sustainability in this setting and replication in other long-term care settings.

It is important to consider the limitations. The evaluation sample size is small and may exhibit a positivity bias. Data collection was anonymous, with only one demographic question on staff position. We thought that anonymity would facilitate candid responses. Notably, staff who completed the survey before the program provided negative evaluations of their relationships with family caregivers. We are planning a more robust evaluation and replication in other settings.

## 6. Conclusions

Before the COVID-19 pandemic raised awareness of family’s critical role in long-term care, the leadership and staff of this long-term care-home believed that families of long-term care residents wanted to continue to offer personal and instrumental care after the people they have cared for at home were admitted to long-term care. We documented the design of this family involvement initiative to educate family caregivers and staff to partner in residents’ care. We explicitly acknowledged and worked with the safety and governance regulations to reintroduce family involvement. The police and background checks, volunteer training, and family caregiver care skills training ensure resident safety. Youville’s Family Involvement Program gives family and family caregivers an explicit role as partners in long-term care. The mixed methods pilot evaluation documented improvements in staff and family relationships. Implementing the design in long-term care was a complex endeavor and we hope that this article encourages further reflection on how to meaningfully involve families and family caregivers in long-term care. The Family Involvement Program continues to thrive in Youville.

## Figures and Tables

**Figure 1 healthcare-12-00523-f001:**
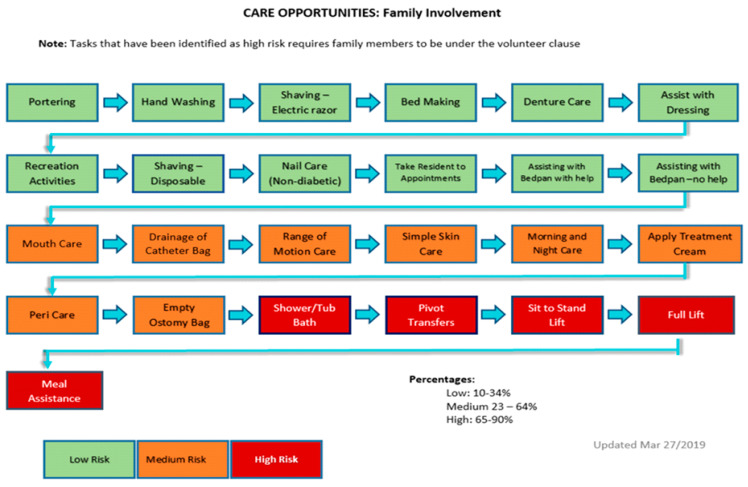
Care opportunities.

**Figure 2 healthcare-12-00523-f002:**
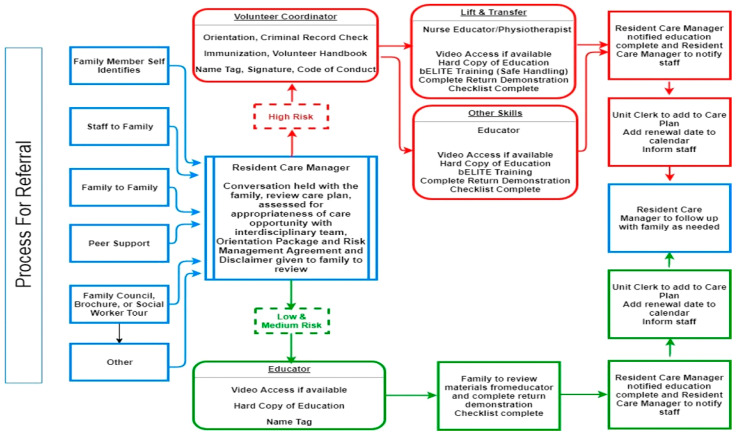
Referral Process.

**Table 1 healthcare-12-00523-t001:** The workplan sequence.

Activity	Date
Sought feedback from family and residents through the Family Council, family, and resident satisfaction surveys.	2017
Conducted survey of staff to determine what they thought of family involvement.	2017
Created “Startup” Steering Committee of leadership, staff, and families.	2017
Facilitated mission discernment to see if project aligned with ethical principles.	2017
Convened the Steering Committee, including Youville Home leadership team, resident families, human resources manager, researchers, and representatives from legal, risk management, best practice, and finance.	2017
Delineated workplan based on recommendations from Steering Committee.	
Established six subcommittees to meet workplan goals and milestones: (1) Policy and Standards, (2) Roles and Responsibilities, (3) Risk Management, (4) Production, (5) Communications, and (6) Sustainability. Each subcommittee was led by the leadership team at Youville Home in 2017.	2017
Applied for funding (Sister’s Legacy Fund).	2018
Hired a coordinator to support the program manager.	2018
Partnered with the Northern Institute of Technology film students to create the instructional videos with the scripts co-designed by the production subcommittee.	2018
Pre-education staff survey (originally 2020, revised because of COVID-19).	2022
Delivered pilot education to staff, families, and residents.	(Revised) 2022

**Table 2 healthcare-12-00523-t002:** Committees and roles.

Committee	Responsibilities
Policy and Standards	Evaluated the consistency of family involvement with policies/standards. Identified where policies might need to be changed to allow family caregivers to be involved. (e.g., policy on use of lifts).
Roles and Responsibilities	Identified and defined stakeholders’ and participants’ roles and responsibilities in the project. Established conflict resolution process.
Risk Management	Created a risk register of the care activities that families expressed an interest in participating in. Assessed the risk incurred for each activity by family, staff, and setting, determined factors that mitigated risks, and assessed the risk when mitigating factors were implemented.
Production	Created scripts and produced instructional videos for the project (training for caregivers).
Communication	Vetted all of the information needed for stakeholders and oversaw production of all communication documents (brochures, process documents, and internal and external marketing).
Sustainability	Oversaw the processes, standards, and strategies designed to sustain the program as part of regular practice at the site.

**Table 3 healthcare-12-00523-t003:** Family caregivers’ perceptions of family involvement program (2023).

Family Caregivers (n = 20)		
	Range	Mean (SD)
How important do you think the FIP is in improving residents’ quality of life?	2 to 5	4.55 (0.76)
How important do you think the FIP is in improving the quality of care?	3 to 5	4.50 (0.60)
Do you think involving yourself in care opportunities is important?	3 to 5	4.53 (0.72)
Do you feel the FIP improves family and staff relationships?	3 to 5	4.21 (0.71)
How satisfied are you with your level of involvement in your family member’s care?	2 to 5	4.22 (0.73)

**Table 4 healthcare-12-00523-t004:** Health providers: comparison of perceptions before (2022) and after (2023) experience in Family Involvement Program.

	Providers 2022 (n = 15)		Providers 2023 (n = 44)		Mann–Whitney
Question	Range	Mean (SD)	Range	Mean (SD)	*p* Sig
How important do you think the Family Involvement Program is in improving residents’ quality of life?	1 to 5	3.25 (1.8)	3 to 5	4.54 (0.5)	0.020
How important do you think the Family Involvement Program is in improving the quality of care?	1 to 5	2.93 (1.4)	3 to 5	4.74 (0.4)	0.010
How satisfied are you with the family involvement education?	1 to 5	2.55 (1.6)	3 to 5	4.45 (0.6)	0.003
How satisfied are you with your working relationship with families?	1 to 5	3.25 (1.5)	3 to 5	4.31 (0.7)	0.039

## Data Availability

Data are available from the authors on request. Email: sdanders@ualberta.ca.

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
