# Peer review of "Family Involvement Training for Staff and Family Caregivers: Case Report on Program Design and Mixed Methods Evaluation"

_healthcare, 2024, doi:10.3390/healthcare12050523_

Round 1
Reviewer 1 Report
Comments and Suggestions for Authors
Dear authors
Congratulations for the research.
The research it is very interesting and the thematic relevant.
Title: OK
Abstract: The methods and the results must be clear in the abstract
Introduction:
The bold in “loved one” (line 44)
Please clarify the meaning of “support due to age” (line 45).
In my opinion is more emotional and financial support than “financial Help”.
Methods
1 – will be relevant report the study design
2 – Provided more information of the context (Youville Home). In my countries, long-term care is an umbrella concept that includes different residential facilities. According to information, the Youville Home is not a permanent resident for older adults (intermediate care ?).
3 – Do the authors have data on the number of healthcare professionals?
4 – According to the ADAPTS Implementation Science Model, the assessment is not clear (what services or resources are needed?)
5 – Table 1 - Sought feedback from s family and residents through the Family Council
6 – sample Size “Three quarters (76%) of respondents in the three groups agreed that 162 strategies or initiatives were needed to involve families”.
7 - Dear authors, you do not use a paired group to evaluate the difference in health providers’ attitudes before and after the education. (Wilcoxon text?). What p-value was considered in the statistical results?
Results.
In lines 302 to 307 the p values were different from table 4.
Discussion.
Will be relevant greater include of your results in the discussion.
The authors reported “Eltaybani and colleagues [22] conclude that family-oriented interventions were associated with high-quality care and better family staff relationships. However, the reference is related to a review protocol.
Limitation: ok
Conclusion: Ok
Organization and style of presentation: needs be edited
Good luck and thank you very much for giving me a chance to review.
Author Response
Thank you so much for the review. When we read the paper now with your review, we are so glad that we have your peer review.
Dear authors
Congratulations for the research.
The research it is very interesting and the thematic relevant.
Yes, we think family involvement in long-term care is important. Covid-19 and dealing with the public health protocols made this challenging but also important. We were shocked by low ratings of their relationships with family caregivers before started the roll out.
Title: OK
We have changed the title based on reviewer 2's suggestions
Abstract: The methods and the results must be clear in the abstract
Thank you, we have completely revised this abstract.
Introduction:
The bold in “loved one” (line 44)
We removed loved one except where participants used it. We recognize that relationships are complex and not necessarily loving.
Please clarify the meaning of “support due to age” (line 45).
Thank you so much, it should be because of frailty.
In my opinion is more emotional and financial support than “financial Help”.
Yes thank you. we have rectified this. In Canada, care costs are included in our universal healthcare, but food and lodging are not. There are also out-of-pocket costs such as transportation.
Methods
1 – will be relevant report the study design
We clarified that the program description is a case report and the then described the evaluation in methods.
2 – Provided more information of the context (Youville Home). In my countries, long-term care is an umbrella concept that includes different residential facilities. According to information, the Youville Home is not a permanent resident for older adults (intermediate care ?).
Thank you, yes Youville is a nursing or long-term care home, residents final home. We described the criteria for admission.
3 – Do the authors have data on the number of healthcare professionals?
Over the course of this research, the staffing model has changed, and all long-term care settings including Youville have been, and continue to be short-staffed. Initially, staffing enabled about 27 minutes of care per day per resident. Provincial funding has increased to enable long-term care settings to hire more healthcare aides. With the increased care complexity, they are trying to fund 4 hours of care per day per resident. However, long-term care settings are competing for a limited number of healthcare aides.
4 – According to the ADAPTS Implementation Science Model, the assessment is not clear (what services or resources are needed?)
This intervention was developed by site administrators and leadership as a side of the desk activity because they wanted to create a better working relationship between families and staff. We have clarified the assessment.
5 – Table 1 - Sought feedback from s family and residents through the Family Council
In Alberta, the policy directs long-term care facilities to have a Family Council. We recruited families and sought family council input.
6 – sample Size “Three quarters (76%) of respondents in the three groups agreed that strategies or initiatives were needed to involve families”.
We have added participant numbers for these consultations.
7 - Dear authors, you do not use a paired group to evaluate the difference in health providers’ attitudes before and after the education. (Wilcoxon text?). What p-value was considered in the statistical results?
We used the Mann- Whitney U test. It is the non-parametric alternative to the t-test for independent samples. It compared medians by converting the scores to ranks across the two groups, then evaluates whether the ranks for the two groups differ. We clarified that we used the .05 as the significance level and Thank you. We should have used the more conservative equal variances, not assumed.
Results.
In lines 302 to 307 the p values were different from table 4.
Thank you, We also used the more conservative p-values and double-checked to make sure we transferred them to the text correctly
Discussion. Will be relevant greater include of your results in the discussion.
The authors reported “Eltaybani and colleagues [22] conclude that family-oriented interventions were associated with high-quality care and better family staff relationships. However, the reference is related to a review protocol.
We have revised the discussion to make it clear to readers that we discuss the design and implementation first, then the evaluation. Yes it should have been Hayward’s not Eltaybani. We should have checked twice.
Limitation: ok
Conclusion: Ok
Organization and style of presentation: needs be edited
We hope that the revisions make it more readable. Good luck and thank you very much for giving me a chance to review.
Thank you so much for your recommendations and the review. We know how long this takes.
Reviewer 2 Report
Comments and Suggestions for Authors
Thank you for the opportunity to review this manuscript. This research study contributes to our understanding of the family caregivers, particularly in the long-term care residence. Here are my insights for improvement:
1. Title
Consider providing the title. It is unclear, invalid meaning, theoretical and content reflection. Moreover, the tile lack of in-depth reflection to the methods.
2. Abstract
The abstract section is unclear what is the introduction lead to exist in the main purpose of the study. Moreover, unclear and too general method using, finding, conclusion, and implications (see line 18-31).
3. Introduction
The introduction is too short in explaining the general problem statement what family caregivers are significant. What gaps of the phenomenon during a COVID-19 pandemic related to long-term care? Moreover, the title does not reflect to the COVID-19, but the introduction section provides the COVID-19 information. Should be clear, why?
What gaps in existing previous studies? How many factors this study provided to investigate? What objectives are needed to be studied? What research question did the authors need to asked and answered the phenomenon as above mentioned (see introduction)?
4. Literature review
The study is unclearly conceptualized the theories/concepts/issues if the authors do not add the review section.
5. Methods
The method section is unclear, what study design is? Where the study setting is? How do the authors provide to recruit the participants? How do the authors analyze the data?
6. Results
The results are invalid evidence supports. The findings are too general information. What are real results from the participants. For instance, quantitative is to short and general statistics. What is results are means? What results are communicated for? How do the results reflect to the title/question/objective.
In qualitative findings, the quotation, coding, and interpretation are not reflected to the participants, theories and concepts (see results section).
7. Discussion
The discussion is too general and does not follow with the main findings. Should be clear what findings are? Should be clear what findings are related to others? Should be clear how findings are theoretical, practical, and policy implications.
8. Conclusion
The conclusion is too short and unclear. This section does not follow with the main summarization of the findings.
Comments on the Quality of English LanguageModerate editing is required. This is because some word, sentence, and paragraph are unrelated to academic writing and scientific logic.
Author Response
Thank you for the opportunity to review this manuscript. This research study contributes to our understanding of the family caregivers, particularly in the long-term care residence. Here are my insights for improvement:
Thank you for this review. While it was sobering, we really appreciate the chance to revise the paper.
- Title
Consider providing the title. It is unclear, invalid meaning, theoretical and content reflection. Moreover, the tile lack of in-depth reflection to the methods.
We have changed the title to reflect the methods.
- Abstract
The abstract section is unclear what is the introduction lead to exist in the main purpose of the study. Moreover, unclear and too general method using, finding, conclusion, and implications (see line 18-31).
Thank you, it was very unclear. We have revised it to make the objectives, methods, findings and conclusion clear.
- Introduction
The introduction is too short in explaining the general problem statement what family caregivers are significant. What gaps of the phenomenon during a COVID-19 pandemic related to long-term care? Moreover, the title does not reflect to the COVID-19, but the introduction section provides the COVID-19 information. Should be clear, why?
What gaps in existing previous studies? How many factors this study provided to investigate? What objectives are needed to be studied? What research question did the authors need to asked and answered the phenomenon as above mentioned (see introduction)?
We realized that the introduction focused too much on COVID. In this study, COVID interfered with what we planned to do, but then it reinforced that family caregivers were important in long-term care.
We have revised the introduction to clarify that we had 2 objectives: a case study report on design and implementation, which fills a gap reported in reviews, and to report on pilot evaluation. Then we focused our literature review on the weaknesses in family involvement interventions detailed in systematic and critical reviews
- Literature review
The study is unclearly conceptualized the theories/concepts/issues if the authors do not add the review section.
c, because administrators in the Youville Home wanted to be more welcoming to families to create a more home-like environment. They were surprised by how time consuming the process was because of long-term care policies and regulations. We hope that this case study article encourages further reflection on the implementation, even for small changes.
- Methods
The method section is unclear, what study design is? Where the study setting is? How do the authors provide to recruit the participants? How do the authors analyze the data?
Again, thank you. We hope that we have used your advice wisely. We report on the case study first and then provided the methods for the evaluation (recruitment, data collection, analysis methods
- Results
The results are invalid evidence supports. The findings are too general information. What are real results from the participants. For instance, quantitative is to short and general statistics. What is results are means? What results are communicated for? How do the results reflect to the title/question/objective.
In qualitative findings, the quotation, coding, and interpretation are not reflected to the participants, theories and concepts (see results section).
Thank you. This is a pilot evaluation, conducted in ongoing COVID outbreaks and staffing shortages. We had intended to do a much more rigorous evaluation. We were shocked at staff’s low ratings of their relationships with family caregivers and heartened by the changes after the Family Involvement education for staff and family caregivers.
- Discussion
The discussion is too general and does not follow with the main findings. Should be clear what findings are? Should be clear what findings are related to others? Should be clear how findings are theoretical, practical, and policy implications.
Thank-you. We structured the discussion into 3 sections: 1) design and implementation, 2) evaluation and 3) strengths and limitations. We begin each paragraph in the discussion with our findings and then discuss our findings in relation to the literature.
- Conclusion
The conclusion is too short and unclear. This section does not follow with the main summarization of the findings.
We have revised the conclusion to reflect and summarize our objectives.
Comments on the Quality of English Language
Moderate editing is required. This is because some word, sentence, and paragraph are unrelated to academic writing and scientific logic.
Thank you for your review. We really appreciate how long it takes to review papers.
Round 2
Reviewer 2 Report
Comments and Suggestions for Authors
Thank you for your revision. As previous suggestion and comment were revised and well-suited publication in the Long-Term Care for Older Adults. Good luck!
Comments on the Quality of English LanguageMinor editing required
Author Response
Thank you so much! We have edited it and hope we have gotten all the small errors.